# PUNR: Pre-training with User Behavior Modeling for News Recommendation

**Guangyuan Ma**[1,2], **Hongtao Liu**[3], **Xing Wu**[1,2],
**Wanhui Qian**[3], **Zhepeng Lv**[3], **Qing Yang**[3], **Songlin Hu**[1,2†]

[1]Institute of Information Engineering, Chinese Academy of Sciences, Beijing, China
[2]School of Cyber Security, University of Chinese Academy of Sciences, Beijing, China
[3]Du Xiaoman Finance, Beijing, China

{maguangyuan,wuxing,husonglin}@iie.ac.cn
{liuhongtao01,qianwanhui,lvzhepeng,yangqing}@duxiaoman.com

## Abstract

News recommendation aims to predict click behaviors based on user behaviors. How to effectively model the user representations is the key to recommending preferred news. Existing works are mostly focused on improvements in the supervised fine-tuning stage. However, there is still a lack of PLM-based unsupervised pre-training methods optimized for user representations. In this work, we propose an unsupervised pre-training paradigm with two tasks, i.e. user behavior masking and user behavior generation, both towards effective user behavior modeling. Firstly, we introduce the user behavior masking pre-training task to recover the masked user behaviors based on their contextual behaviors. In this way, the model could capture a much stronger and more comprehensive user news reading pattern. Besides, we incorporate a novel auxiliary user behavior generation pre-training task to enhance the user representation vector derived from the user encoder. We use the above pre-trained user modeling encoder to obtain news and user representations in downstream fine-tuning. Evaluations on the real-world news benchmark show significant performance improvements over existing baselines.

## 1 Introduction

News recommendation is an essential technology for predicting candidate news via the history of user behaviors. It is widely used in online news feeds and platforms to model user behaviors and push personalized news to their users.

Existing news recommendation methods often employ Pre-trained Language Models (PLMs) to produce news vectors and user vectors, and match personalized candidate news based on the dot product distances of these two vectors. For example, NRMS-BERT (Wu et al., 2019c, 2021) replaces multi-head self-attention with pre-trained PLM, such as BERT (Devlin et al., 2019), RoBERTa (Liu et al., 2019) and UniLM (Dong et al., 2019), to

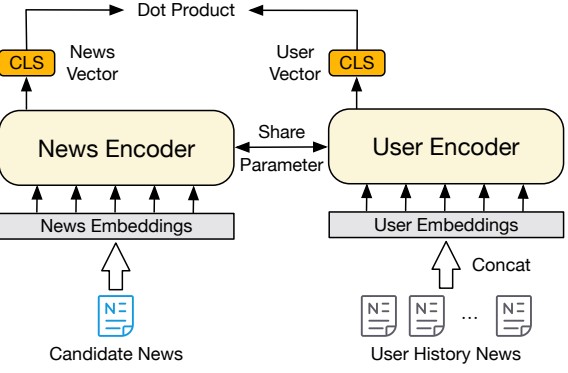

Figure 1: The architecture of our user modeling method for new recommendations in downstream fine-tuning. A Siamese encoder is used for both modeling the news and user embeddings. *Note that we use the textual titles of user-clicked news as inputs in pre-training, rather than a pure news id.*

get news embeddings of user behaviors. Then a hierarchical shallow user encoder is used on top of the news encoder, in order to aggregate the user behaviors to get a user representation. Such dual-tower PLM-based user modeling architecture is widely adopted in current works of news recommendations. However, the separated forward of the individual user behaviors (i.e. users' clicked news titles) and hierarchical aggregation result in potentially weak user modeling ability and complicated model architectures.

On the contrary, UNBERT (Zhang et al., 2021b) proposes to concatenate the candidate news and user behavior news together, and predict the matching score via a classification head with the word-level vector and the news-level vector. AMM (Zhang et al., 2021a) concatenates the cross pairs of title, abstract, and body together between the users' history news and candidate news, to extract multi-field matching representation in both within-field and cross-field. Such single-tower classification techniques result in good user-candidate behavior modeling but are relatively computationally expen-

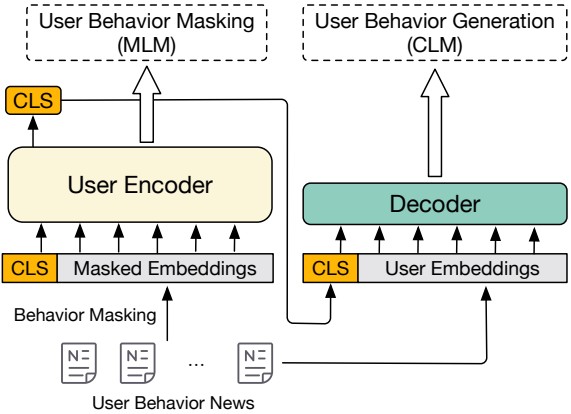

Figure 2: Pre-training design of our proposed method, PUNR. We incorporate a special user behavior masking strategy and a user behavior generation task in the pre-training stage of the user encoder.

sive. In order to achieve joint modeling of user behaviors while maintaining a balanced computational offload, as is shown in Figure 1, we simplify the architecture of the user encoder and only use a Siamese encoder for modeling both news vectors and user vectors. The concatenation of user history news brings better alignment of user behaviors. And we refer to this joint modeling of users' clicked histories as *User Modeling*.

Effectively recommending content to users relies heavily on the ability to accurately model their behaviors (Sun et al., 2019). PTUM (Wu et al., 2020a) proposes pre-training multi-head self-attention layers from scratch to predict masked behaviors and the next K behaviors. However, PTUM only utilizes a small amount of in-domain data and does not leverage the power of PLMs. Unfortunately, pre-training for news recommendation has not yet been extensively explored, as most existing methods focus on improving the fine-tuning stage by adapting an off-the-shelf PLM as the news encoder. We argue that pre-training a PLM specifically tailored for news recommendation is essential for achieving optimal performance. Moreover, research on modeling user behaviors for news recommendations is still in its early stages.

In our work, we propose **PUNR**, Pre-training with User behavior modeling for News Recommendation, seeking to incorporate unsupervised pre-training signals for better user modeling ability. On top of the Siamese architecture, we incorporate two pre-training tasks tailored for user modeling. Detailed designs are shown in Figure 2.

• **User behavior masking task.** We randomly

mask the whole behavior spans in the inputs of the user encoder. The task for the user encoder is to recover the masked behavior spans with its contextual user history news texts. The modeling ability of the user encoder is improved and makes stronger recommending performance.

Inspired by recent representation generative pre-training technology in IR community (Lu et al., 2021; Wu et al., 2022), the information can be compressed into the representation vector through a generative pre-training task with a shallow Transformer decoder. This auto-regression pre-training schema enlights us to explore new pre-training paradigms for better modeling a user vector in news recommendations.

• **User behavior generation task.** A single-layer auto-regression Transformers decoder is added on top of the user encoder. The decoder takes the user vector and user history news as input. The task for the decoder is to generate the whole user history news in an auto-regression way assisted by the user vector. This auto-regression pretask enables a more powerful user representation learning.

To the best of our knowledge, we make the first effort to bring unsupervised pre-training tasks to the PLM-based news recommender system. The joint use of user behavior generation and user behavior masking design for user vector optimization makes a steady recommendation improvement. We also propose a simple yet effective Siamese architecture for encoding both the news vectors and user vectors. Experiment results on the real-world news recommendation task show considerable performance gains over multiple baselines. In addition, we conduct several ablation studies to verify the robustness of our method.

Our contribution can be summarized as follows:
• We propose an effective user modeling Siamese encoder architecture for both encoding the news and user representations.
• We incorporate user behavior generation and user behavior masking tasks into the unsupervised pre-training of the PLM-based news recommendation system.
• Experiments show that our work achieves considerable performance gains over multiple baselines.

## 2   Approach

In this section, we describe the problem statement of news recommendation and the detailed imple-

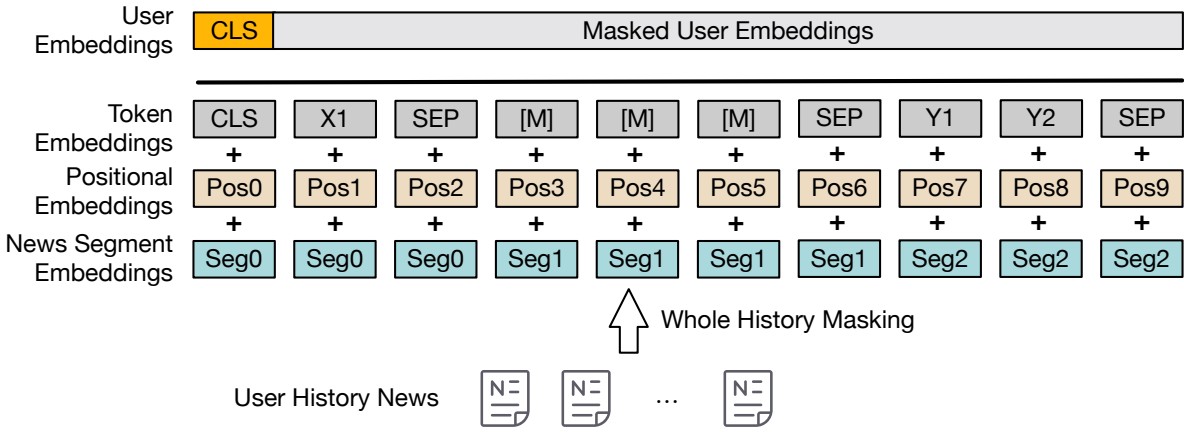

Figure 3: The implementation of user model embeddings for PUNR. The title fields of the clicked news are concatenated together for joint modeling of each user's behavior. The joint user modeling not only simplifies the architecture of the user encoder to a single BERT model but also enables stronger recommendation effects.

mentations of the pre-training and fine-tuning.

## 2.1 Problem Statement

Given a user with a set of clicked news $\{D_1^u, D_2^u, ..., D_{|U|}^u\} \in \mathcal{U}$ (denoted as behaviors), the task for news recommendation is to distinguish preferable (positive) news from candidate sets $\{D_1^v, D_2^v, ..., D_{|V|}^v\} \in \mathcal{V}$ based on user's behaviors.

## 2.2 Pre-training with User Behavior Modeling

We employ a Transformers-based Pretrained Language Model (PLM), e.g. BERT, to serve as a user encoder to generate the representation of the user's behaviors. Specifically, given a clicked news set $\mathcal{U}$, we tokenize each news input[1] and concatenate them together as a line of input tokens $\mathbb{T} = \{t_0, t_1, ..., t_n\}$ to produce token embeddings $E_{token}$. Following (Zhang et al., 2021b), we employ positional embeddings $E_{pos}$ and news segment embeddings $E_{seg}$ to represent positional and segmenting information. The input embeddings $E$ is the sum of the above three embeddings, as is shown in Figure 3.

$$E(\mathbb{T}) = E_{token} + E_{pos} + E_{seg} \quad (1)$$

**User Behavior Masking** During the pre-training stage, we employ a user span masking strategy for modeling the user behaviors with token-level Mask Language Modeling (MLM) task. Given a total mask ratio $\alpha$, we especially use [MASK] tokens to replace the whole piece of the news segment.

The percentage of mask tokens used by user behavior masking is $\beta$. And the rest of the tokens are randomly masked. We denote the set of masked tokens as $\mathbb{M}$.

The masked text $m(\mathbb{T})$ is firstly converted to user embeddings $E(m(\mathbb{T}))$, and then forwarded through the L-layer Transformers-based user encoder ($Enc$) to recover all masked positions with the MLM loss. The loss is formulated as:

$$\mathcal{L}_{mlm} = -\sum_{i \in \mathbb{M}} \log p(t_i | Enc(E(m(\mathbb{T})))) \quad (2)$$

During the recovery of user behavior masking, the user model predicts the entirely masked behaviors with unmasked or partially masked contexts. This masking strategy will bring consistent user behavior alignment in user modeling.

**User Behavior Generation** For each layer $l \in \{1, ..., L\}$ of the user encoder, the output hidden states are denoted as follows:

$$\mathbf{H}^l = \{\mathbf{h}_0^l, \mathbf{h}_1^l, ..., \mathbf{h}_N^l\} \quad (3)$$

Here we simply use the last hidden states of the [CLS] position, i.e. $h_0^{last}$ as the user vector. Based on the concatenated input of user vector CLS and user embeddings, a single-layer Transformers-based auto-regression decoder ($Dec$) is introduced to perform the generation task of the entire user behavior texts.

$$\mathcal{L}_{dec} = -\sum_{i \in \mathbb{T}} \log p(t_{i+1} | Dec(h_0^{last}, E(\mathbb{T}))) \quad (4)$$

---

[1] Following previous works, we use the title of news as inputs.

Similar to the discussion in (Lu et al., 2021; Wu et al., 2022), the single-layer decoder has limited modeling capacity. So it needs to rely on the information from the user vector to finish the generation of entire textual user behaviors. Thus the token-level behavior information is compressed into the user vector, resulting in stronger user behavior modeling.

The pre-training loss is the sum of the encoder MLM loss and the decoder CLM loss.

$$\mathcal{L} = \mathcal{L}_{mlm} + \mathcal{L}_{dec} \qquad (5)$$

## 2.3 Fine-tuning

Fine-tuning is conducted on news recommendation benchmarks for verifying the effectiveness of our work. For simplicity of downstream design, we share the parameters between the news encoder and the user encoder, for both modeling the candidate news and the user's clicked news. The encoder is initialized from the above pre-trained PLM. The clicked news is jointly forwarded through the user encoder to obtain the user vector $u$ at the CLS position. And the candidate news is separately forwarded through the news encoder to obtain the individual news vectors $v$. Following NRMS-BERT (Wu et al., 2021), we use the negative sampling strategy, utilizing a cross-entropy loss for minimizing the distance between the user vector $u$ and the positive news vector $v^+$ while pushing away the negative news vectors $v^-$.

$$\mathcal{L}_{\text{ft}} = -\log \frac{\exp(d(u, v^+)))}{\sum \exp((d(u, v^+) + d(u, v^-)))} \quad (6)$$

where $d$ means the dot product operation.

## 3 Experiments

In this section, we introduce the experimental settings of our work, including the pre-training, fine-tuning, and baseline methods. Then we analyze the main experimental results.

## 3.1 Pre-training

**Datasets** The MIND dataset (Wu et al., 2020b) is used as the pre-training corpus. We use 2.2 million click behaviors from the training set of MIND for the pre-training of our model. The target max length is set to 512.

Since the auto-regression decoder is not initialized, we use the original BERT (Devlin et al., 2019)

pre-training corpus Wiki+BookCorpus for the initialization of the decoder. It contains about 5.6 million documents. NLTK sentence tokenizer is used to split the corpus and dynamically padded to the target max length.

**Implementation** The user encoder in the pre-training stage is initialized from the pre-trained BERT-base checkpoint. We first freeze the BERT encoder and only pre-train the single-layer auto-regression with general Wiki+BookCorpus. The AdamW optimizer is used with a batch size of 512, a learning rate of 3e-4, and pre-training steps of 150k for the full initialization of the decoder on the general corpus. Then we pre-train the full model on MIND corpus using AdamW optimizer with a batch size of 256, a learning rate of 1e-5, and pre-training steps of 10k. A linear scheduler with a warmup ratio of 0.1 is used for the learning rate schedule. We set the total masking ratio $\alpha$ to 0.3 and the user behavior masking ratio in the total masked token sets $\beta$ to 0.3 because this combination brings the best results. We use 8 Tesla V100 GPUs and takes about 1 day to finish the whole pre-training. We vary the pre-training seeds and take an average of 3 scores as our main results.

## 3.2 Fine-tuning

**Datasets** The MIND benchmark (Wu et al., 2020b) is used to evaluate the effectiveness of our method. It is a read-world English news recommendation dataset, which contains 160k news texts, 1 million users, 15.8 million impressions, and 24 million click behaviors collected from Microsoft News in six weeks. MIND has two versions of datasets with different sizes. MIND-large includes 2.2 million samples in the training set and 365K samples in the development set. MIND-small is a randomly sampled version of the whole dataset, which contains 50k users and their behavior logs from the MIND dataset.

**Implementation** The pre-trained encoder is used to initialize the downstream news and user encoders. The parameters are shared across these two encoders because we found that using a Siamese encoder improves performance. We fine-tune the encoder using AdamW optimizer with a batch size of 128, and a learning rate of 3e-5 for 3 epochs. The linear scheduler with a warmup ratio of 0.1 is also used in the fine-tuning stage. Following (Wu et al., 2021), 1 positive and 4 negative news are randomly sampled from the candidate sets. The

| Methods | MIND-Small | | | | MIND-Large | | | |
|---|---|---|---|---|---|---|---|---|
| | AUC | MRR | nDCG@5 | nDCG@10 | AUC | MRR | nDCG@5 | nDCG@10 |
| LibFM | 59.74 | 26.33 | 27.95 | 34.29 | 61.85 | 29.45 | 31.45 | 37.13 |
| DeepFM | 59.89 | 26.21 | 27.74 | 34.06 | 61.87 | 29.3 | 31.35 | 37.05 |
| DKN | 61.75 | 27.05 | 28.9 | 35.38 | 64.07 | 30.42 | 32.92 | 38.66 |
| NPA | 63.21 | 29.11 | 31.7 | 37.81 | 65.92 | 32.07 | 34.72 | 40.37 |
| NAML | 65.5 | 30.39 | 33.08 | 39.31 | 66.46 | 32.75 | 35.66 | 41.4 |
| LSTUR | 64.38 | 29.46 | 31.89 | 38.17 | 67.08 | 32.36 | 35.15 | 40.93 |
| NRMS | 64.83 | 30.01 | 32.52 | 38.92 | 67.66 | 33.25 | 36.28 | 41.98 |
| FIM | 65.02 | 30.26 | 32.91 | 39.1 | 67.87 | 33.46 | 36.53 | 42.21 |
| NRMS-BERT | 65.52† | 31.00† | 33.87† | 40.38† | 69.5 | 34.75 | 37.99 | 43.72 |
| UNBERT | 67.62 | 31.72 | 34.75 | 41.02 | 70.68 | **35.68** | **39.13** | 44.78 |
| AMM | 67.96 | 32.98 | 36.64 | 42.77 | - | - | - | - |
| **PUNR** | $\mathbf{68.89}_{\pm 0.17}{}^{*}$ | $\mathbf{33.33}_{\pm 0.08}{}^{*}$ | $\mathbf{36.94}_{\pm 0.11}{}^{*}$ | $\mathbf{43.10}_{\pm 0.09}{}^{*}$ | $\mathbf{71.03}_{\pm 0.04}{}^{*}$ | $35.17_{\pm 0.05}$ | $39.04_{\pm 0.04}$ | $\mathbf{45.41}_{\pm 0.05}{}^{*}$ |

Table 1: Main results on MIND datasets. The best scores are marked in bold. $^{*}$Two-tailed t-tests demonstrate statistically significant improvements in our method over baselines ( p-value $\leq 0.05$ ). †The scores are based on our reproduction.

max number of user behaviors for the user encoder is set to 50, and the max length for each news title is set to 30. The seed for fine-tuning is fixed to 42 for reproducibility.

**Evaluation Metrics** Following (Wu et al., 2020b), several evaluation metrics, including AUC, MRR, nDCG@5, and nDCG@10, are used for evaluating the performance. The scores are averaged across all impression logs. We train the model on MIND training sets and report the results on MIND development sets.

## 3.3 Baselines

We compare the performance of our model with multiple baselines, including general TF-IDF (Ramos et al., 2003) feature-based methods, convolutional neural network (CNN-based) methods, deep neural network (DNN-based) methods, and PLM-based methods. Most of the experiment results come from (Zhang et al., 2021b), while results of the PLM-based methods come from their original papers.
**LibFM** (Rendle, 2012) utilizes TF-IDF features of users' clicked news and candidate news in factorization machines.
**DeepFM** (Guo et al., 2017) also uses TF-IDF features in deep factorization machines.
**DKN** (Wang et al., 2018) uses information from a knowledge graph and CNN networks for news feature extraction.
**NPA** (Wu et al., 2019b) utilizes the CNN network for news feature extracting and personalized attention network incorporating user IDs.

**NAML** (Wu et al., 2019a) applies the neural news recommendation with attentive multi-view learning for news representation extraction.
**LSTUR** (An et al., 2019) applies the neural news recommendation with long- and short-term user representations for news recommendation.
**NRMS** (Wu et al., 2019c) utilizes multi-head self-attention for news recommendation.
**FIM** (Wang et al., 2020) employs a fine-grained interest matching for neural news recommendation.
**NRMS-BERT** (Wu et al., 2021) is a dual-tower PLM-based neural news recommendation method that uses PLM, such as BERT, for news feature extraction.
**UNBERT** (Zhang et al., 2021b) is a single-tower PLM-based neural news recommendation method that concatenates candidate news and users' clicked news together and utilizes word-level and news-level features for click probability prediction.
**AMM** (Zhang et al., 2021a) is a single-tower PLM-based neural news recommendation method that concatenates the cross pairs of title, abstract, and body together from candidate news and users' clicked news to extract multifield matching representation for click probability prediction.

## 3.4 Main Results

The main results are presented in Table 1. Scores on MIND-Small and MIND-Large show a significant improvement over multiple baselines, including recent PLM-based neural recommendation methods like NRMS-BERT, UNBERT, and AMM. Our method completely beats the previous dual-tower methods NRMS-BERT. When comparing

|  | AUC | MRR | nDCG@5 | nDCG@10 |
|---|---|---|---|---|
| NRMS-BERT (Baseline) | 69.5 | 34.75 | 37.99 | 43.72 |
| w/ User Modeling | 70.52 (+1.02) | 34.60 | 38.38 (+0.39) | 44.81 (+1.09) |
| w/ Only User Behavior Masking | 70.79 (+1.29) | 34.86 (+0.11) | 38.65 (+0.66) | 45.11 (+1.39) |
| w/ Only User Behavior Generation | 70.92 (+1.42) | 35.02 (+0.27) | 38.86 (+0.87) | 45.28 (+1.56) |
| w/ Both Pre-training Tasks | 71.07 (+1.57) | 35.19 (+0.44) | 39.07 (+1.08) | 45.45 (+1.73) |

Table 2: Ablation study for user modeling and different pre-training tasks on MIND-Large benchmarks. Performance gains compared to the NRMS-BERT baseline are marked in red.

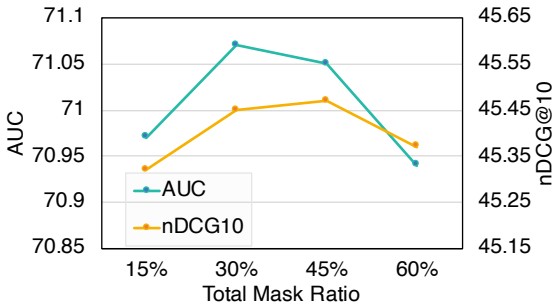

Figure 4: Performances with different total mask ratios $\alpha$ on MIND-Large Benchmark.

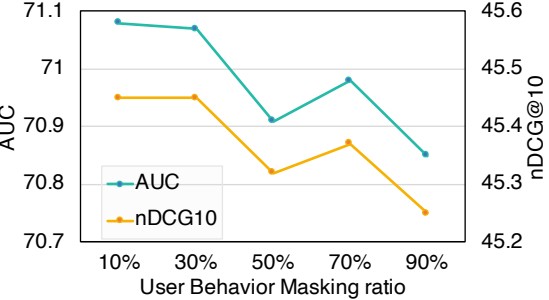

Figure 5: Performances with different user behavior masking ratios $\beta$ on MIND-Large Benchmark.

with the single-tower baseline, although AMM utilizes multifield matching information with a single-tower encoder for click probability prediction, our dual-tower method still outperforms it by +0.93 on AUC, +0.35 on MRR, +0.3 on nDCG@5, +0.33 on nDCG@10 of MIND-Small benchmark. Our method outperforms UNBERT by +0.35 on AUC and +0.63 on nDCG@10.

These performance gains could be attributed to our special user modeling design with user behavior masking and user behavior generation tasks. These tasks are tailored for user modeling and user vector pre-training, thus bringing continuous unsupervised signals for effective user representation learning. We will perform a detailed analysis in the next section.

## 4 Analysis

Given the performance gains of our user behavior pre-training methods, we systematically analyze the following scientific questions in this section. **Q1.** How does user modeling contribute to the performances? **Q2.** How do user behavior masking and user behavior generation contribute to the performances? **Q3.** How do total mask ratio $\alpha$ and behavior mask ratio $\beta$ influence the performances? **Q4.** Is the initialization of the auto-

regression decoder necessary for the pre-training? **Q5.** How does performance change when choosing a Siamese or fully separated encoder in fine-tuning? **Q6.** Will different pooling methods affect the performances?

Detailed analyses (**A**) are conducted below one by one. The seed for all experiments in this section is fixed to 42 for reproducibility.

### 4.1 Influence of User Modeling and User Behavior Pre-training Tasks

Table 2 shows the ablation results for user modeling and user behavior pre-training tasks.

**A1.** NRMS-BERT is a typical PLM-based news recommendation baseline method. After we simplify the architecture of the user encoder and switch to the Siamese encoder for user modeling fine-tuning, our model achieves +1.02 points on AUC over the previous baseline.

We attribute this increment to better user modeling in the user encoder. The concatenation of user-clicked histories brings token-level deep interaction within Transformers blocks, thus bringing joint modeling and better alignment to the user representation.

| AR-Dec | AUC | MRR | nDCG@5 | nDCG@10 |
|---|---|---|---|---|
| Pre-trained | 71.07 | 35.19 | 39.07 | 45.45 |
| Random init | 70.43 | 34.68 | 38.48 | 44.93 |

Table 3: Ablation study for the initialization of Auto-Regression Decoder (AR-Dec) with general corpus on MIND-Large benchmarks.

| | AUC | MRR | nDCG@5 | nDCG@10 |
|---|---|---|---|---|
| Siamese | 71.07 | 35.19 | 39.07 | 45.45 |
| Separated | 70.32 | 34.61 | 38.37 | 44.84 |

Table 4: Ablation study for the Siamese or separated encoder in fine-tuning on MIND-Large benchmarks.

**A2.** Besides the user modeling, we discuss the impact of different pre-training tasks. After incorporating the user behavior masking and user behavior generation tasks into the pre-training stage, our model achieves +1.29 and +1.42 on AUC compared to the baseline. The usage of both two tasks achieves +1.57 on AUC, which demonstrates the effectiveness of both pre-training tasks.

As discussed in Section 2, the user behavior masking task encourages the model to predict the entirely masked textual span with other unmasked contextual information, thus bringing a stronger signal for user behavior modeling. The user behavior generation task utilizes a single-layer auto-regression decoder to fully generate the entire user behaviors based on the user vector, thus enriching the ability of the user representation during the backpropagation update from the decoder layer.

### 4.2 Influence of the Total Mask Ratio and User Behavior Masking Ratio

**A3.** Performances with different total mask ratios are shown in Figure 4. We fix the user behavior mask ratio $\beta$ at 30% and change the total mask ratio between 15% and 60%. We empirically found that our pre-training method can tolerate a wide range of total mask ratios, and slightly increasing it to 30% or 45% will be helpful to achieve higher performance.

The results "w/ Only User Behavior Generation" in Table 2 show that a suitable user behavior masking is helpful for higher performance. Thus we also adjust the user behavior masking ratio $\beta$ to test a suitable setting. The total mask ratio is fixed at 30%. We found that keeping the user behavior masking ratio less than 30% achieves higher performance. We believe that making the user behavior

| | AUC | MRR | nDCG@5 | nDCG@10 |
|---|---|---|---|---|
| CLS | 71.07 | 35.19 | 39.07 | 45.45 |
| Average | 70.95 | 35.03 | 38.92 | 45.28 |
| Attention | 70.77 | 34.92 | 38.79 | 45.20 |

Table 5: Ablation study for different pooling methods on MIND-Large benchmarks.

masking ratio too high will cause a much more challenging task, thus unsuitable for the reconstruction of masked behaviors.

### 4.3 Impact for the Initialization of the Auto-Regression Decoder

**A4.** Since the single-layer Auto-Regression Decoder (AR-Dec) can not be initialized from a pre-trained checkpoint, we freeze the encoder and specifically train the AR-Dec with the same general pre-training corpus as BERT. We compare the pre-trained and random initialized AR-Dec in Table 3. Results show that the decoder needs a proper initialization on the general corpus. A randomly started decoder will corrupt the further pre-training with user behavior generation task.

### 4.4 Impact of Siamese or Fully Separated Encoder

**A5.** Performances of using a Siamese or fully separated encoder for fine-tuning is presented in Table 4. Similar to the observation in the IR community (Dong et al., 2022), sharing parameters between the news and user encoder brings better alignment of news and user vectors, thus achieving higher performance.

### 4.5 Impact of Different Pooling Methods

**A6.** Performances with different pooling methods are listed in Table 5. Following (Wu et al., 2021), three different pooling methods are compared in our works, including **1) CLS**: Directly using representation at CLS position as user vector, which is widely used to obtain a sentence embedding. We use this method in our main results. **2) Average**: Using the average of hidden states from the last layer. **3) Attention**: Using a parameterized MLP weighting network to aggregate a sentence embedding from the hidden states of the last layer.

Different from previous works (Wu et al., 2021), our method achieves higher scores on parameter-free pooling methods, e.g. CLS and Average, but has a lower performance on Attention pooling

methods. We believe that our pre-training method with user behavior generation is tailored for direct user representation learning. And a parameterized pooling method will weaken the effort of this pre-training process.

# 5 Related Works

In this section, we introduce traditional neural news recommendation systems and PLM-based neural news recommendation systems.

## 5.1 Neural News Recommendations

News recommendation systems aim to predict users' preferred news based on their browsing histories. Early manual feature-based recommendation systems (Rendle, 2012; Guo et al., 2017) often employ deep factorization machines for click-through rate prediction. In recent years, traditional news recommendation systems utilize neural networks, e.g. convolutional neural network (CNN) and deep neural network (DNN), for neural-based news and user feature extraction. DKN (Wang et al., 2018) utilizes knowledge-graph-enhanced CNN networks for news feature extraction. NPA (Wu et al., 2019b) also utilizes CNN networks with personalized attention networks incorporating user IDs. NAML (Wu et al., 2019a) is a classical DNN-based neural news recommendation with attentive multi-view learning for news representation learning. LSTUR (An et al., 2019) applies long and short-term user representations into DNN-based neural news recommendation. NRMS (Wu et al., 2019c) is another classical DNN-based neural news recommendation method that utilizes multi-head self-attention. FIM (Wang et al., 2020) incorporates a fine-grained interest matching into DNN-based methods. Traditional neural news recommendation systems are mostly lightweight networks, designed for online inference of news recommendations.

## 5.2 PLM-based News Recommendations

The bloom of Pre-trained Language Models (PLMs), such as BERT (Devlin et al., 2019), RoBERTa (Liu et al., 2019) and UniLM (Dong et al., 2019), invigorates the development of PLM-based neural recommendation systems for higher recommendation performances. With sufficient unsupervised pre-training corpus, PLMs are powerful for representation extraction. Thus they are used as the backbone of PLM-based news recommendation systems. PLM-NR (Wu et al., 2021) designs a dual-tower architecture that utilizes PLM for news embedding extraction. A hierarchical user encoder is used for aggregating the user embeddings. The distances between the news embeddings and the user embeddings are calculated with dot product or cosine similarities. PLM-NR works as a classical PLM-based news recommendation baseline. Similar to PLM-NR, we also use a dual-tower architecture for inferencing. But we use concatenated user clicked histories, rather than separated encoding, for effective user modeling.

UNBERT (Zhang et al., 2021b) proposes to concatenate candidate news and users' clicked news together and use a single-tower BERT encoder for direct click probability prediction. It aggregates word-level and news-level features and achieves competing recommendation performances. AMM (Zhang et al., 2021a) pushes the steps further and proposes to concatenate the cross pairs of title, abstract, and body together from candidate news and users' clicked news. Thus it extracts multifield matching representation for click probability prediction. Similar to the above methods, we jointly model the user-clicked behaviors. But we do not concatenate the candidates with the users' behaviors, because our work focuses on user representation optimization. Our work remains a dual-tower architecture for balancing effectiveness and efficiency. Furthermore, to the best of our knowledge, we are the first to bring unsupervised pre-training tasks tailored for user modeling into the pre-training stage of the user model. Our pre-training methods bring better modeling ability to the user encoder, which is proven to be effective with extensive experiments.

# 6 Conclusion

This paper proposes to pre-train with user behavior modeling for better news recommendations. A simple yet efficient user modeling encoder is used for better alignment of user behaviors. On top of this, we further incorporate the user behavior masking task and the user behavior generation task for effective unsupervised pre-training. Experiments on real-world news benchmarks show that our works significantly improve the performances of PLM-based news recommendations.

# Limitations

As our pre-training method is tailored for PLM-based news recommendations, it requires relatively

larger GPU resources than non-PLM-based methods. Moreover, the joint modeling of the user encoder brings increased performance, but the news embeddings from user behaviors cannot be preencoded and thus uncacheable. This hinders fast inferencing in low-resource scenarios. We will investigate the effect of downscaling PLMs or distillation for faster inferencing, and leave this to further works.

## Ethics Statement

The authors declare that they have no conflict of interest.

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
