# OpenReview forum: "PUNR: Pre-training with User Behavior Modeling for News Recommendation"
_EMNLP/2023/Conference — EMNLP 2023 Findings_

### Official Review · Reviewer_ed5M · 2023-07-19

**Typos Grammar Style And Presentation Improvements:** There should be some punctuations aft…
**Soundness:** 4

**Excitement:**

4: Strong: This paper deepens the understanding of some phenomenon or lowers the barriers to an existing research direction.

**Missing References:**

None

**Paper Topic And Main Contributions:**

This research proposes an unsupervised pre-training method for news recommendation based on user behaviors. The authors introduce two tasks: user behavior masking to recover masked behaviors and user behavior generation to enhance user representations. Their approach outperforms existing methods in real-world news recommendation benchmarks.

**Questions For The Authors:**

1. Why are  some methods mentioned in related works especially those for PLM-based news recommendation not compared in experiments?

**Reasons To Accept:**

1. The problem studied in this paper is interesting and important, but rarely researched. Thus, this research has some novelty.

2. The proposed pretraining methods, i.e., user behavior masking to recover masked behaviors and user behavior generation to enhance user representations, are reasonable, although they are standard techniques in language modeling.

3. The experiments are extensive and solid. The improvement brought by the proposed method is significant.

**Reasons To Reject:**

1. There are a few writing flaws. For example, there should be some punctuations after the equations.

2. Besides the two MIND datasets, more datasets could be used in experiments.

**Reproducibility:**

4: Could mostly reproduce the results, but there may be some variation because of sample variance or minor variations in their interpretation of the protocol or method.

**Reviewer Confidence:**

4: Quite sure. I tried to check the important points carefully. It's unlikely, though conceivable, that I missed something that should affect my ratings.

---

> ### Author Rebuttal · Authors · 2023-08-27
>
> Thank you very much for your insightful comments. We will answer your questions as follows.
>
> **Q1.** About writing flaws and more datasets.
>
> **A1.** Thank you for your suggestions. We will correct them immediately and include more datasets in our extended works.
>
> **Q2.** Methods mentioned in related works are not compared in experiments.
>
> **A2.** Note that NRMS-BERT is actually from PLM-NR [1], which has already been compared in our works. We choose NRMS-BERT for a fair comparison. Nevertheless, we will include more related latest baselines in our next version. Thanks again for your suggestions.
>
> [1] Empowering News Recommendation with Pre-trained Language Models. SIGIR 2021

---

### Official Review · Reviewer_YPna · 2023-08-03

**Soundness:** 2

**Excitement:**

2: Mediocre: This paper makes marginal contributions (vs non-contemporaneous work), so I would rather not see it in the conference.

**Paper Topic And Main Contributions:**

This paper introduces a pretraining model PUNR for news recommendation. PUNR is pretrained by two objectives of BERT-like masked behavior modeling and GPT-like autoregressive behavior generation. Comprehensive experiments show that PUNR beats several news recommendation baselines.

**Reasons To Accept:**

This paper explores user behavior pretraining, which is less studied by previous research. The idea is generally well stated and experiments are thorough.

**Reasons To Reject:**

1. The major weakness of this paper is the overlap of pretraining and finetuning data scopes. In principle, pretraining and finetuning data should not be overlapped. Otherwise, data contamination [1] makes the pretrained model memorize the data and cannot be well generalized to various downstream tasks. The essence (or target) of PLM is to pretrain on large-scale unlabeled text corpus and finetune on various downstream tasks. For example, GPT-3 and T5 [2,3] are pretrained on Common Crawl, WebText and Wikipedia corpus and finetuned on downstream supervised tasks. Pretraining on task-specific data makes the model difficult to be adapted to other tasks. The authors pretrain the model on MIND and also finetune on MIND, which is irrational and should not be called pertaining (it can be regarded as typical training), unless the author tests the model on other datasets or tasks. Pretraining and finetuning on the same dataset MIND is not a good setting for PLM study. Especially note that MIND-small is a subset of MIND-large, the MIND-small test set is also included in pertaining MIND-large corpus.

2. As the authors use relatively high-cost computation resources to "pretrain" the model of large parameters, advanced news recommendation baselines should be compared, including [4,5,6,7]. Fairly speaking, the chosen baselines are not updated to advanced news recommendation works of the recent two years.

3. Equation 4 is not right where $t_{<i}$ is either missing or not self-consistent with the notation of $E(T)$.

4. Minor concern of mine: Masked user behavior modeling is intuitive. However, incorporating GPT-like user behavior generation by introducing an additional decoder seems redundant and unnecessary, though the authors claim with experiments that it can help to learn the representation of user vectors. Since the user behavior and its containing news articles are represented in a hierarchy, I also doubt that it is too difficult for the decoder to generate such hierarchical and complex user behaviors only based on a user representation vector. Considering that the user representation vector [CLS] is a one-dimension vector (not like the $n\times d$ hidden states of Transformer or T5 encoder outputs), I doubt if this vector can generate the whole complex user behaviors.

5. Initializing the generative decoder with the original BERT is irrational. Moreover, using BERT's architecture to model generation is not a good choice. BERT's architecture is designed and pretrained for Mask Language Modeling, and the authors revised it to do autoregressive Language Modeling (CLM in the paper). The pretraining gap between Mask Language Modeling and Language Modeling exists.


[1] Data Contamination: From Memorization to Exploitation, on ACL-2022

[2] Language Models are Few-Shot Learners, on NIPS-2020

[3] Exploring the Limits of Transfer Learning with a Unified Text-to-Text Transformer, on JMLR-2020

[4] HieRec: Hierarchical User Interest Modeling for Personalized News Recommendation, on ACL-2021

[5] MINER: Multi-Interest Matching Network for News Recommendation, on ACL-2022

[6] MTRec: Multi-Task Learning over BERT for News Recommendation, on ACL-2022

[7] DIGAT: DIGAT: Modeling News Recommendation with Dual-Graph Interaction, on EMNLP-2022

**Reproducibility:**

2: Would be hard pressed to reproduce the results. The contribution depends on data that are simply not available outside the author's institution or consortium; not enough details are provided.

**Reviewer Confidence:**

4: Quite sure. I tried to check the important points carefully. It's unlikely, though conceivable, that I missed something that should affect my ratings.

**Typos Grammar Style And Presentation Improvements:**

In line 005, 'works are mostly focused on' should be 'works mostly focus on'.

The terminology "auto-regression" should be "auto-regressive" which is standard used in our NLP community.

---

> ### Author Rebuttal · Authors · 2023-08-27
>
> Thank you very much for your insightful comments. We will answer your questions as follows.
>
> **Q1.** Data Contamination issue.
>
> **A1.** Thanks for your very detailed insights.
>
> 1) We believe that our work does not have such a data contamination issue. In paper [1], it pre-trains with both training corpus and **human labels** in downstream test sets. However, we only pre-train with 2.2 million click behaviors from the training set of MIND, which does not contain any labels from impression logs.
> 2) Our work can be viewed as in-domain pre-training techniques. In-domain pre-training is actually prevalent in NLP communities. For example, GPT-3 is pre-trained with Wikipedia and Web pages. Popular downstream tasks like Natural Questions [2] and TriviaQA [3] are also made from Wikipedia. A popular pre-training architecture in the IR community, named Condenser [4], is pre-trained with MS-MARCO web collections [5]. It also fine-tuned and tested on MS-MARCO and TREC [6] benchmarks, which are all related to MS-MARCO collections.
> 3) Overall, we think that the data scopes in pretraining and fine-tuning could overlap, especially in the in-domain tasks, under the premise of ensuring that the labels do not leak.
>
> [1] Data Contamination: From Memorization to Exploitation, ACL-2022
>
> [2] Natural Questions: A Benchmark for Question Answering Research. Trans. Assoc. Comput. Linguistics 7: 452-466 (2019)
>
> [3] TriviaQA: A Large Scale Distantly Supervised Challenge Dataset for Reading Comprehension. ACL (1) 2017: 1601-1611
>
> [4] Condenser: a Pre-training Architecture for Dense Retrieval. EMNLP (1) 2021: 981-993
>
> [5] MS MARCO: A Human Generated MAchine Reading COmprehension Dataset. CoCo@NIPS 2016
>
> [6] Overview of the TREC 2019 deep learning track. CoRR abs/2003.07820 (2020)
>
> **Q2.** Advanced news recommendation baselines should be compared.
>
> **A2.** Thanks for your suggestions.
>
> 1) Our work focuses on the pre-training stage, which is orthogonal to fine-tuned methods. Compared with advanced fine-tuned recommendation architectures may not be a very fair practice. For example, HieRec uses hierarchical user interest matching. MINER uses multi-interest user modeling with multi-field information. MTRec uses additive attention and gradient surgery.
> 2) Our architecture merely uses a Siamese encoder (one BERT model) as both the user and news encoder for recommendation. It's simple and uses fewer parameters than the above methods.
>
> We will consider exploring combining our pre-training methods with advanced fine-tuning architecture in our extended works.
>
>
> **Q3.** Equation 4 is not right.
>
> **A3.** Thanks for pointing this out. We will correct this in our next versions.
>
> **Q4.** It is too difficult for the decoder to generate such hierarchical and complex user behaviors only based on a user representation vector.
>
> **A4.** Yes, this task is difficult. We believe that besides relying on the self-attention of GPT, this difficult task will encourage the GPT to rely more on the information within the [CLS] vector to finish the generation of user behaviors. In this way, we seek to "compress" the information of user behaviors into the [CLS] vector. Thus the enhanced representations will do better in downstream recommendations.
>
> As recorded in our pre-training logs, The final MLM loss for the BERT encoder is 1.8739, and the final CLM loss for the decoder is 1.6354. Thus the decoder is actually capable of finishing such a generation task.
>
> **Q5.** Initializing the generative decoder with the original BERT is irrational.
>
> **A5.** Actually, we do not initialize the generative decoder from BERT. Because there is no off-the-shelf single-layer GPT for initialization of the decoder, as described in the Experiments Section 3.1, we first pre-train the decoder with the Wiki+Bookcorpus, the same pre-training corpus as BERT. Then we use our proposed structure for pre-training with in-domain corpus MIND. Note that we discard the decoder in downstream fine-tuning, as it is only designed for the pre-training stage. We can view the decoder as an assistant for CLS representation of the encoder to learn better user behavior modeling ability.

---

### Official Review · Reviewer_n3WE · 2023-08-05

**Soundness:** 3

**Excitement:**

3: Ambivalent: It has merits (e.g., it reports state-of-the-art results, the idea is nice), but there are key weaknesses (e.g., it describes incremental work), and it can significantly benefit from another round of revision. However, I won't object to accepting it if my co-reviewers champion it.

**Paper Topic And Main Contributions:**

The authors propose an architecture that includes news and user representation using pre-training, which incorporate user behavior generation and user behavior masking tasks, to solve news recommendation.

**Questions For The Authors:**

1. Can we look at case studies to see why this structure is more efficient than other baselines?
2. What is the effect of word representations using random initialization?


**Reasons To Accept:**

1. Experiments fully prove the effectiveness of the proposed method.
2. The description is logical and easy to follow.
3. The proposed architecture is straightforward and easy to understand.


**Reasons To Reject:**

1. The idea of user behavior pre-training is very common in recommendation systems.
2. According to the results in Table 2, the improvement of the model effect is gradual. Also, why not use UNBERT as a baseline for comparison, in Table 2?

**Reproducibility:**

4: Could mostly reproduce the results, but there may be some variation because of sample variance or minor variations in their interpretation of the protocol or method.

**Reviewer Confidence:**

4: Quite sure. I tried to check the important points carefully. It's unlikely, though conceivable, that I missed something that should affect my ratings.

---

> ### Author Rebuttal · Authors · 2023-08-27
>
> Thank you very much for your insightful comments. We will answer your questions as follows.
>
> **Q1.** The idea of user behavior pre-training is very common in recommendation systems.
>
> **A1.** Yes, user behavior modeling is prevalent in recommendation systems. However, we argue that:
> 1) Previous studies often focus on training with user IDs [1, 2]. Different from previous studies, our work focuses on pre-training with textual behavior sequences to enhance the user representations, i.e. [CLS] from the user encoder.
> 2) We mark our core contributions and effectiveness as designing User Behavior Masking and User Behavior Generation tasks tailored for pre-training stronger user representations.
>
> [1] PTUM: Pre-training User Model from Unlabeled User Behaviors via Self-supervision. EMNLP (Findings) 2020
>
> [2] U-BERT: Pre-training User Representations for Improved Recommendation. AAAI 2021
>
> **Q2.** According to the results in Table 2, the improvement of the model effect is gradual. Also, why not use UNBERT as a baseline for comparison, in Table 2?
>
> **A2.** Table 2 is the **ablation studies** for investigating the effects of each proposed pre-training task. The model effect is shown gradually for disassembling the performance gains and verifying that our proposed pre-training tasks all contribute to higher performances.
>
> We choose NRMS-BERT as baselines because we derive from a dual-encoder recommendation architecture, e.g. NRMS-BERT, which first generates news vectors and user vectors and then ranks the candidate news with scores of dot product or cosine similarities.
>
> In contrast, UNBERT is a single-encoder architecture that concatenates each candidate news and each user sequence and forwards the PLM with all combinations. UNBERT has a deep interaction of the candidate news and user sequences with self-attention, which is **not a fair comparison baseline** for our ablation studies.
>
> **Q3.** Can we look at case studies to see why this structure is more efficient than other baselines?
>
> **A3.** Here we present the case study as follows. The historical clicked news is listed in Table 1. The news recommendation task aims to rank the candidate news with historical clicked news (Or User Behaviors). Here we show the top-3 results for our method and NRMS-BERT and mark the correct recommendation in bold text.
>
> As is shown in Table 2, our method recommends more diverse news with categories ranging in "Travel", "Lifestyle" and "Food and Drink", and returns the correct news as the second result. In contrast, in Table 3, NRMS-BERT biases the top-3 results all towards the "Sports" category. We contribute our efficient recommendation as special pre-training tasks tailored to enhancing user modeling abilities.
>
> **Table 1.** Historical Clicked News of Impression ID: 96, User ID: U118952.
> | Category                            | Title                                                                                                |
> |-------------------------------------|------------------------------------------------------------------------------------------------------|
> | News                                | Analysis: A Mike Pence Presidency Is No Longer Just Dinner Party Chatter in DC. Is He Up to the Job? |
> | News                                | Analysis: Democrats' impeachment gamble paying off in court of public opinion for now.               |
> | Sports                              | Las Vegas prepares to welcome Raiders, but is it a bad bet?                                          |
> | Travel                              | 19 of the best ski resorts in North America that don't cost a fortune.                               |
> | Sports                              | 3 ways to fix the NFL's onside kick problem.                                                         |
> | Food and Drink                      | Panera Bread worker fired after TikTok exposed frozen mac and cheese.                                |
> | Finance                             | President Trump's trillion-dollar hit to homeowners.                                                 |
> | TV                                  | 'Wheel Of Fortune' Guest Delivers Hilarious, Off The Rails Introduction.                             |
>
> **Table 2.** Recommended by Our Methods. The title marked in **bold** is the ground truth of this example.
> | Category                            | Title                                                                                                |
> |-------------------------------------|------------------------------------------------------------------------------------------------------|
> | Travel                              | Unwanted tourist types: The kinds of travelers becoming more unwelcome                               |
> | Lifestyle                           | **Why I'm Done Celebrating The Accomplishments Of Couples**  |
> | Food and Drink                      | The Real Reason McDonald's Keeps the Filet-O-Fish on Their Menu                                      |
>
> **Table 3.** Recommended by NRMS-BERT
> | Category                            | Title                                                                                                |
> |-------------------------------------|------------------------------------------------------------------------------------------------------|
> | Sports                              | Wilson deletes tweet of him standing over concussed Smith-Schuster                                   |
> | Sports                              | Warriors on pace for worst drop in winning percentage in NBA history                                 |
> | Sports                              | LeBron James on he and Tom Brady: 'We're gonna play until we can't walk'                             |
>
>
> **Q4.** What is the effect of word representations using random initialization?
>
> **A4.** Pre-training with user behavior generation needs an initialized single-layer decoder. Because we don't have a single-layer GPT to initialize this decoder, as described in Section Experiment, we first train it with Wiki+BookCorpus (which is also the BERT pre-training corpus). Random initialization in Table 3 means we omit this step, verifying that this will hurt the performances.

---

### Meta-Review · Area_Chair_3LSR · 2023-09-19

**Recommendation:** 3

**Metareview:**

This paper introduces a pretraining model PUNR for news recommendation. The proposed PUNR is pretrained by two objectives of BERT-like masked behavior modeling and GPT-like autoregressive behavior generation.


Pros:
The proposed paper is backed by thorough experiments, demonstrating its effectiveness. The straightforward and understandable architecture of the proposed method simplifies its comprehension. Furthermore, the pretraining methods are considered reasonable and well-reasoned, aligning with established practices in language modeling.

Cons:
Despite these strengths, there are some concerns raised by the reviewers. The novelty of this paper is relatively limited.  Furthermore, the gradual improvement observed in the experimental results has been identified as a drawback.
Concerns about data overlap between pretraining and finetuning data are significant, with suggestions that this could lead to data contamination and impact the generalization of the model. There are also recommendations to compare the proposed method with recent news recommendation baselines, address equation inconsistencies, evaluate the necessity of an additional decoder, rectify minor writing flaws, and consider using a more diverse set of datasets in experiments to enhance the paper's overall robustness.

---

### Decision · Program_Chairs · 2023-10-07

**Decision:**

Accept-Findings

**Comment:**

This paper introduces a pretraining model PUNR for news recommendation. The proposed PUNR is pretrained by two objectives of BERT-like masked behavior modeling and GPT-like autoregressive behavior generation.


Pros:
The proposed paper is backed by thorough experiments, demonstrating its effectiveness. The straightforward and understandable architecture of the proposed method simplifies its comprehension. Furthermore, the pretraining methods are considered reasonable and well-reasoned, aligning with established practices in language modeling.

Cons:
Despite these strengths, there are some concerns raised by the reviewers. The novelty of this paper is relatively limited.  Furthermore, the gradual improvement observed in the experimental results has been identified as a drawback.
Concerns about data overlap between pretraining and finetuning data are significant, with suggestions that this could lead to data contamination and impact the generalization of the model. There are also recommendations to compare the proposed method with recent news recommendation baselines, address equation inconsistencies, evaluate the necessity of an additional decoder, rectify minor writing flaws, and consider using a more diverse set of datasets in experiments to enhance the paper's overall robustness.